# Vaginal microbiome of women with adenomyosis: A case-control study

**Jitsupa Kunaseth**[1], **Wanwisa Waiyaput**[2], **Prangwalai Chanchaem**[3],
**Vorthon Sawaswong**[3,4], **Rattana Permpech**[5], **Sunchai Payungporn**[3,6‡*],
**Areepan Sophonsritsuk**[1‡*]

**1** Reproductive Endocrinology and Infertility Unit, Department of Obstetrics and Gynaecology, Faculty of Medicine, Ramathibodi Hospital, Mahidol University, Bangkok, Thailand, **2** Office of Research Academic and Innovation, Faculty of Medicine Ramathibodi Hospital, Mahidol University, Bangkok, Thailand, **3** Research Unit of Systems Microbiology, Chulalongkorn University, Bangkok, Thailand, **4** Program in Bioinformatics and Computational Biology, Graduate School, Chulalongkorn University, Bangkok, Thailand, **5** Perioperative Nursing Division, Department of Ramathibodi Nursing Service, Faculty of Medicine Ramathibodi Hospital, Mahidol University, Bangkok, Thailand, **6** Department of Biochemistry, Faculty of Medicine, Chulalongkorn University, Bangkok, Thailand

‡ SP and AS are contributed equally to this work as co-corresponding authors.
* areepan.sop@mahidol.ac.th (AS); sp.medbiochemcu@gmail.com (SP)

**Data Availability Statement:** All relevant data are available in the paper, its Supporting Information files, and on Figshare (DOI: 10.6084/m9.figshare. 18739562).

## Abstract

Immune dysregulation can involve invasion and survival of endometrial glands inside the myometrium of the adenomyosis. There is limited available data concerning alterations of the bacterial microbiome in the reproductive tract of adenomyosis women. The present cross-sectional age-matched study aims to compare vaginal microbiota between women with and without adenomyosis. We recruited women with adenomyosis (N = 40) and age-matched women without adenomyosis (N = 40) from the Departments of Obstetrics and Gynaecology, Ramathibodi Hospital Mahidol University, from August 2020 to January 2021. Vaginal swab samples were collected from the participants. DNA isolation and bacterial 16s rDNA gene sequencing and data analyses were then performed. Comparison of the diversity of vaginal microbiota, microbiota composition, and the operational taxonomic unit (OTU) between adenomyosis and non-adenomyosis (control) groups were undertaken. Data from 40 and 38 women with and without adenomyosis, respectively, were analyzed. Alpha-diversity analysis (Chao1 index) at the species level showed higher vaginal microbial richness in the adenomyosis group when compared with the control group ($p = 0.006$). The linear discriminant analysis effect size technique (LeFSe) indicated an elevated abundance of several vaginal microbial taxa in the adenomyosis group, including *Alloscardovia*, Oscillospirales, Ruminoccoccaceae, *UCG_002*, Oscillospiraceae, *Enhydrobacter*, *Megamonas*, Moraxellaceae, *Subdoligranulum*, Selenomonadaceae, and *Faecalibacterium*. On the other hand, an increase in the abundance of *Megaspehera*, *Fastidiosipila*, Hungateiclostridiaceae, and Clostridia was identified in the control group. Vaginal community state type (CST)-III and -IV were dominated in adenomyosis, while only CST-IV was dominated in the non-adenomyosis group. *Lactobacillus* was the most abundant vaginal microbial in both groups. In this study, the differences in vaginal microbiome profile were noted between adenomyosis and non-adenomyosis group. The increasing of microbial richness was

**Funding:** The study was financially supported by the Faculty of Medicine, Ramathibodi Hospital, Mahidol University (project number RF_63085). The funders had no role in study design, data collection and analysis, decision to publish, or preparation of the manuscript.

**Competing interests:** The authors have declared that no competing interests exist.

associated with adenomyosis. Nevertheless, further investigations were required to elucidate the mechanisms and apply them for clinical implications.

## Background

Adenomyosis is a benign uterine myometrium lesion that is commonly found in women of reproductive age. Most women with adenomyosis present with a wide range of symptoms, such as heavy menstruation, progressive dysmenorrhea, and a decline in fertility [1]. The pathologies defined with endometrial glands and stroma presented within the uterine musculature are surrounded by hyperplastic and hypertrophic smooth muscle [2]. Among several proposed etiological theories for adenomyosis, the most accepted ones include direct invagination of endometrium through the junctional zone (JZ), metaplasia of the Mullerian remnant, and/or displacement of the retrograde endometrium and stem cell into the serosa and myometrium of the uterus [3].

Data from previous studies demonstrate that a vicious cycle of immune dysregulation exists involving invasion and survival of endometrial glands inside the myometrium of the adenomyosis [4, 5]. An alteration of immune factors within the eutopic endometrium was proposed to play a role in the invasion mechanism [5]. Elevation of immunosuppressive cytokines, including interleukin (IL)-10, was observed to be a survival mechanism of the ectopic endometrium in both ectopic and eutopic endometrial glands of adenomyosis [6]. This observation is linked to the survival mechanism of endometrial glands within the myometrium without eradication by the immune system. However, the precise immune dysfunction contributing to adenomyosis development is not completely understood.

A microbiome is a microbial community that occupies a specific habitat and has distinct physicochemical properties within this environment [7]. The microbiome in the human body undergoes a complex interaction between the human host and the community of microorganisms [8]. Previous studies of microbes used culture-dependent techniques, which have limited microorganism detection. In the past decade, the emergence of high-performance DNA sequencing techniques, such as next-generation sequencing (NGS), allows for a comprehensive analysis of the microbiome to be done and provides a broad picture of the bacterial component of the host [9, 10]. Several studies have shown an association between the alteration of microbiota and local or even distance diseases. For example, changes in the intestinal microbiome may contribute to inflammatory bowel disease, obesity, and autoimmune diseases [11].

Over 250 species of vaginal bacteria have been discovered using high-performance DNA sequencing technologies. The vaginal microbiome is a dynamic complex ecosystem that is dominated by *Lactobacillus* spp. with a vaginal pH < 4.5 [12]. Among the 20 species of *Lactobacillus* that have been detected in the vagina, it is common for women to have their vaginal microbiome composed of largely one lactobacillus species [13]. Common *Lactobacillus* species include *L. crispatus*, *L. iners*, *L. jensenii*, and *L. gasseri*. Unique *Lactobacilli* use different mechanisms to maintain homeostasis and protect against vaginal pathogens. The protection is accomplished by creating an unfriendly microenvironment for pathogens. Vaginal epithelium produces glycogen under the influence of progesterone and estrogen [14], which *Lactobacilli* then metabolize to D- and L-lactic acid to maintain a low pH environment [15]. *Lactobacilli* also competitively consume nutrients on the vaginal epithelium [14]. Furthermore, specific *Lactobacillus* spp. produce hydrogen peroxide ($H_2O_2$) and antimicrobial compounds resulting

in inhibiting the growth of pathogens. The healthy vaginal flora combines numerous bacterial species with a certain large proportion of *Lactobacilli*, and other microbes include *Atopobium*, *Anaerococcus*, *Corynebacterium*, *Peptoniphilus*, *Mobiluncus*, *Prevotella*, *Gardnerella*, and *Sneathia* [12, 16]. Alteration of vaginal microbiota can be influenced by many circumstances, such as antibiotics use, hormone treatment, vaginal hygiene, and sexual activity [17–19]. As mentioned above, disturbance of the vaginal microbiome can lead to an unfavorable milieu that reduces preventive properties against pathogen colonization [14, 20, 21].

Previous studies have shown that vaginal microbiota contributes to the immunity against pathogenic bacteria, parasites, and even the human immunodeficiency virus (HIV) virus [21, 22]. In addition, an association between alterations in the vaginal microbiome and reproductive tract infections, including bacterial vaginitis [20, 23], pelvic inflammatory disease [20, 24], and chronic endometritis [24], have also been reported. Recent studies have focused on genomic analysis of vaginal microbiota for adenomyosis and endometriosis have also been reported [25–28]. Moreover, several studies have reported links between vaginal microbiota in other female reproductive diseases [29–35]. Although previous studies reported alterations of the bacterial microbiome in the reproductive tract of adenomyosis women, two of the studies did not directly focus on the comparison of vaginal microbiota between women with and without adenomyosis [25, 27, 36]. One study mainly focused on the microbiota along with the reproductive site [25], and the other study investigated vaginal microbiota and chronic pelvic pain (CPP) associated with and without adenomyosis/endometriosis [36].

The aim of the current study was to provide insight into the relationship between genital tract microbiome and adenomyosis by comparing vaginal microbiota between women with and without this disease. In this study, we analyzed microbiota collected from vaginas of reproductive-aged women with and without adenomyosis using the NGS technique. While some vaginal microbiota may provide protection against pathogens and regulate vaginal homeostasis, a change of certain vaginal microbiota may introduce chronic inflammation that leads to adenomyosis.

## Materials and methods

This age-matched case-control study was conducted from August 2020 to January 2021 in the Division of Reproductive Endocrinology and Infertility, Department of Obstetrics and Gynecology, Faculty of Medicine Ramathibodi Hospital, Mahidol University. This study was approved by the Ethical Clearance Committee on Human Rights Related Research Involving Human Subjects and Faculty of Medicine at Ramathibodi Hospital, Mahidol University (MURA2020/528). Women diagnosed with adenomyosis based on at least three diagnostic ultrasound-based criteria were recruited into the adenomyosis group. The diagnostic criteria included globular shape uterus, asymmetric myometrial wall, thickening of the endometrial-myometrial junction, hyperechoic striae along the sub-endometrial region, and presence of sub-endometrial microcyst. Age-matched (± 1 year) healthy women who visited the outpatient clinic for annual screening with confirmation for normal uterus and adnexa based on clinical examination and ultrasonographic imaging were recruited. Written informed consent was obtained for every participants before the initiation of the study.

### Sample collection

To avoid contamination, vaginal swab samples were collected from subjects before undertaking any vaginal procedures, including pelvic examination and transvaginal ultrasound. To collect the sample, initially, a sterile speculum was inserted into the vagina. The disposable vaginal brush was used to collect vaginal discharge from the upper vagina, after which the

vaginal sample was placed in a sterile test tube containing DNA/RNA Shield™ reagent (ZYMO Research, USA).

## DNA extraction

Genomic DNA was isolated from the vaginal discharge sample using the GenUP™ gDNA Kit (Biotechrabbit, Germany). DNA extraction included lysis, transfer of the supernatant to a new tube followed by addition of binding buffer, applying the mixture to a mini-filter and eluting DNA with elution buffer. Adequate amounts of extracted DNA were ensured with concentrations > 20 ng/μl [37].

## Amplification of bacterial 16S rDNA

The bacterial 16S rDNA gene was amplified. First, the V3-V4 region of the 16S rDNA gene was amplified by polymerase chain reaction (PCR). The phasing primer set consisted of the targeted primer sequences: (1) 515F:5′-TGCCAGCMGCCGCGGTAA-3′ and (2) 806R: 5′-GGACTACHVGGGTWTCTAAT-3′. The PCR reaction in 20 μl total volume included 20 ng of DNA template, 0.2 μM of each primer, 0.2 mM of dNTPs, 1× Phusion green HF Buffer, and 0.4 U of Phusion DNA polymerase (Thermo Scientific, USA). The amplification was conducted for 30 s at 98˚C to denature the DNA, followed by 25 three-step temperature cycles of 98˚C for 10 s, 53˚C for 25 s, and 72˚C for 10 min. For the final step, the elongation process was performed for 10 min at 72˚C before obtaining 290 bp amplicons [37].

## Sequencing of bacterial 16S rDNA and library preparation

Before the sequencing step, the amplified products from the first round of polymerase chain reaction (PCR) were re-amplified by Illumina sequencing primers, multiplexing indexes, and Illumina adaptors. The PCR products after the re-amplification step were incorporated with Illumina sequencing adaptors and indices. The amplified genes were then separated by 2% agarose gel electrophoresis. The selected bands were cut and purified using QIA quick Gel Extraction Kit (QIAGEN, Germany). Subsequently, library quantification was carried out by real-time polymerase chain reaction (qPCR) using Illumina KAPA library quantification kits (Kapa Biosystems, USA). Finally, the library was normalized and pooled for a final concentration of 2 nM. Library quality control was also performed in which low-quality sample data were excluded from further analyses to ensure data validity. MiSeq v2 reagent kit on MiSeq platform (Illumina, USA) was used for the DNA sequencing step. Ten picomoles of the library with 20% spike-in PhiX were loaded. Finally, library sequencing was repeatedly performed for paired-end 2 × 250 cycles.

## Data processing and statistical analysis

Raw sequencing data were demultiplexed by MiSeq reporter software (version 2.6.2.3). The FASTQ files were analyzed using the QIIME2 pipeline (version 2021.4) [38]. The Phred quality score (Q30) was then used to merge and filter the paired-end reads. Next, the merged reads were de-duplicated and clustered with 97% similarity by VSEARCH [39]. The chimeric sequences were filtered out using a UCHIME algorithm [40]. Finally, the filtered reads were classified based on 99% OTUs clustered 16S Silva Database version 138 [41] using the VSEARCH algorithm [39]. The OTUs of *Lactobacillus* were separately classified by the sklearn classifier [42] against the manually curated reference sequences of *Lactobacillus* species. A vaginal bacterial diversity comparison, including α-diversity, β-diversity, rarefaction curve, and statistical analyses, were evaluated using plug-in implemented for QIIME2 software. For α-

diversity, Chao1 index and Shannon index were calculated for operational taxonomic unit (OTU) richness comparison and OTU evenness with richness comparison, respectively. Microbial abundance differentiation was compared using the linear discriminant analysis (LDA) effective size (LEfSe) [43]. The significant difference of taxa ($p$-value < 0.05) with an LDA score > 2 were shown as a cladogram plot [37].

Vaginal bacterial communities were categorized into five clusters, community-state type (CST), based on the group with the largest ratio of reads according to Ravel et al.'s study [44]. A vaginal sample with the proportion of the most abundant taxa less than 30% is not designated (No type) [45].

Continuous variables were defined as mean ± standard deviation (SD) for normal distribution and median ± interquartile range (IQR) for non-normal distribution variables. Categorical variables were presented as number and percent. Demographic data among both groups were compared using the chi-square and Mann-Whitney U tests. The number of samples in CSTs was analyzed by the chi-square. IBM SPSS for Windows version 25 (IBM Corp., Armonk, NY, USA) was used for statistical analysis. All differences were considered significant at a $p$-value < 0.05.

## Results

The present study recruited 40 women with adenomyosis and 40 women without adenomyosis. Participants from both groups were not significantly different in terms of demographic backgrounds, including age, body mass index (BMI), menstrual phase, and parity (Table 1). Among women with adenomyosis, 80% of those (N = 32) were diagnosed with diffuse-type in which most subjects presented with pelvic pain or abnormal bleeding (S1 Table). The vaginal samples were collected from all participants; however, two samples were excluded from the control group due to poor DNA quality after library quality check. Therefore, 78 samples were used in the subsequent analysis (Fig 1).

Next, the vaginal microbiota was analyzed from 78 specimens using 16s rDNA sequencing techniques. Overall, a total of 2,236,457 reads were acquired from those samples with 290 bp per read with a median of 16,353 reads for each sample (min–max, 3,029–110,351 reads per sample). The reads were saturated for bacterial community evaluation with sufficient depth, which showed a high abundance of microbial diversity described by the alpha rarefaction curve (S1 Fig). Here, 83 OTUs were observed from both groups. Medians and interquartile ranges (IQRs) of the number of OTUs identified in each sample were 25.00 and 10.50 and 21.00 and 8.75 in the adenomyosis and control groups, respectively.

**Table 1. Participants' demographic data.**

| Characteristics | Adenomyosis (n = 40) | Control (n = 40) | *p*-value |
|---|---|---|---|
| Age, years (mean ± SD) | 42.7 ± 5.7 | 42.4 ± 5.0 | 0.305 |
| BMI, kg/m$^2$ (mean ± SD) | 24.2 ± 4.2 | 24.8 ± 5.3 | 0.566 |
| Menstrual cycle phase, N (%) | | | |
| • Follicular | 17 (42.5) | 12 (30.0) | 0.509 |
| • Luteal | 14 (35.0) | 17 (42.5) | |
| • Undeterminable | 9 (22.5) | 11 (27.5) | |
| Parity N (%) | | | |
| • Nullipara | 21 (52.5) | 18 (45.0) | 0.532 |
| • Multipara | 19 (47.5) | 22 (55.0) | |

Note: BMI, Body mass index; SD, Standard deviation.

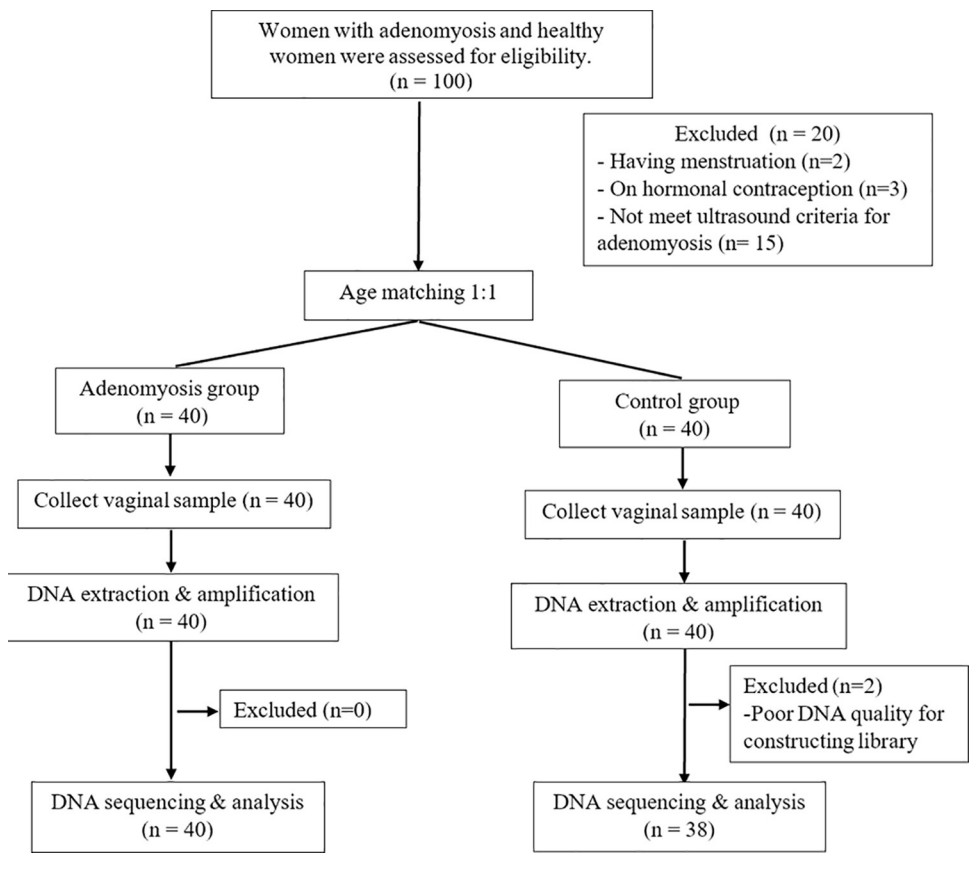

**Fig 1. Study process.**

## Microbial diversity between adenomyosis and control groups

Alpha diversity analysis (Chao1 index) at species level showed higher vaginal microbial richness in the adenomyosis group when compared with the control group ($p = 0.006$). However, microbial diversity was not different between the two groups based on the Shannon index ($p = 0.734$) (Fig 2). Beta diversity analysis of vaginal microbiota was performed using both the

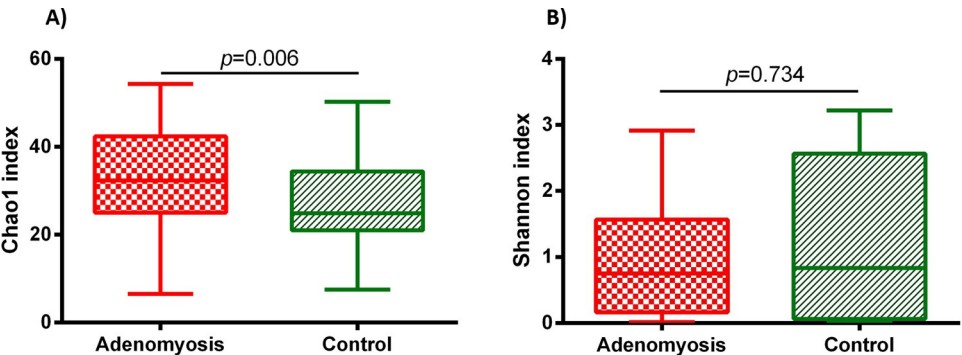

**Fig 2.** Boxplot showing alpha diversity of vaginal microbial in women with and without adenomyosis by (A) Chao1 (richness) and (B) Shannon index (microbial diversity).

Bray-Curtis distance and Jaccard index were not significantly different between women with and without adenomyosis (S2 Fig).

## Relative taxonomy abundance between adenomyosis and control groups

The taxa from eight phyla were obtained from vaginal samples of women with adenomyosis and control groups. The result showed that Firmicutes was the most abundant phylum found in both groups, followed by Actinobacteria, Bacteroidota, Fusobacteria, Patescibacteria, Proteobacteria, Campilobacteria, and Verrucomicrobia (Fig 3A). After considering the genus classification scale, 32 genera from both groups were identified. In this case, *Lactobacillus* was dominant in both groups without significant abundance different (51.20% and 50.54% in adenomyosis and control groups, respectively, $p > 0.05$) (Fig 3B). A large proportion of *Gardnerella* was taken by both groups, with a larger proportion in the adenomyosis group. In terms of

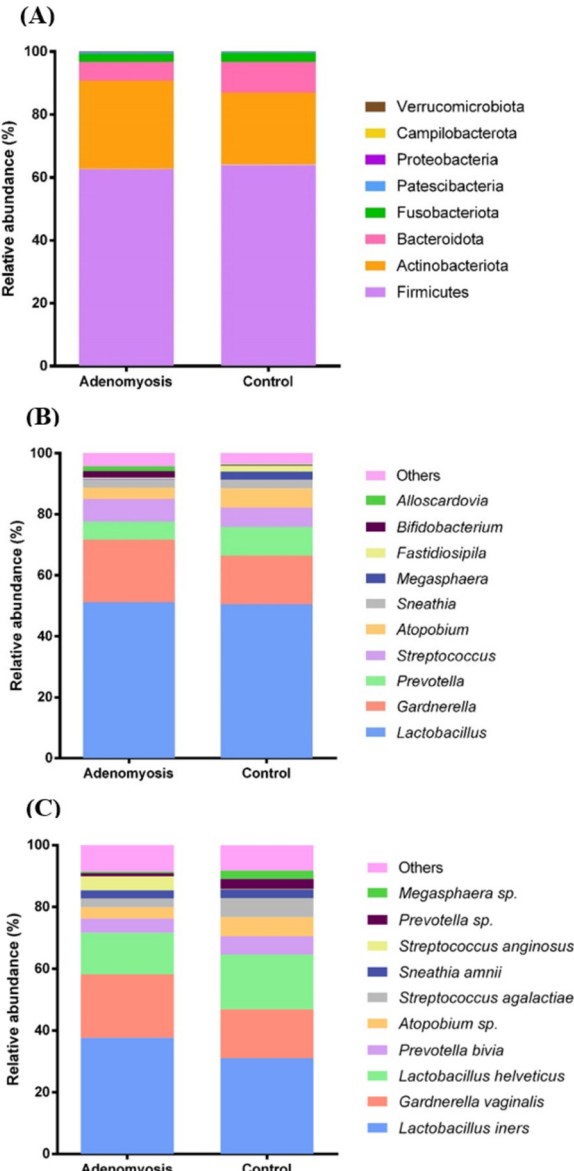

**Fig 3.** Taxonomy bar charts of vaginal microbiota at (A) phylum level and (B) genus level (C) species level.

species level, *L. iners* shared the largest proportion in both groups, with a higher proportion in the adenomyosis group. *G. vaginalis* was detected in the second large proportion in both groups (Fig 3C).

## Differential analysis of taxonomy profiles

To identify the presence of abundant differential taxa of the vaginal microbiota between the adenomyosis and control groups, the linear discriminant analysis effect size technique (LEfSe) was performed. The results highlighted the elevated abundance of several vaginal microbial taxa in the adenomyosis group, including genus *Alloscardovia*, order Oscillospirales, family Ruminoccoccaceae, genus *UCG_002*, family Oscillospiraceae, genus *Enhydrobacter*, genus *Megamonas*, family Moraxellaceae, genus *Subdoligranulum*, family Selenomonadaceae, and genus *Faecalibacterium*. On the other hand, an increase in the abundance of genus *Megaspehera*, genus *Fastidiosipila*, family Hungateiclostridiaceae, and order Clostridia were identified in the control group. The relationship among significantly enriched microorganisms was demonstrated using a cladogram (Fig 4).

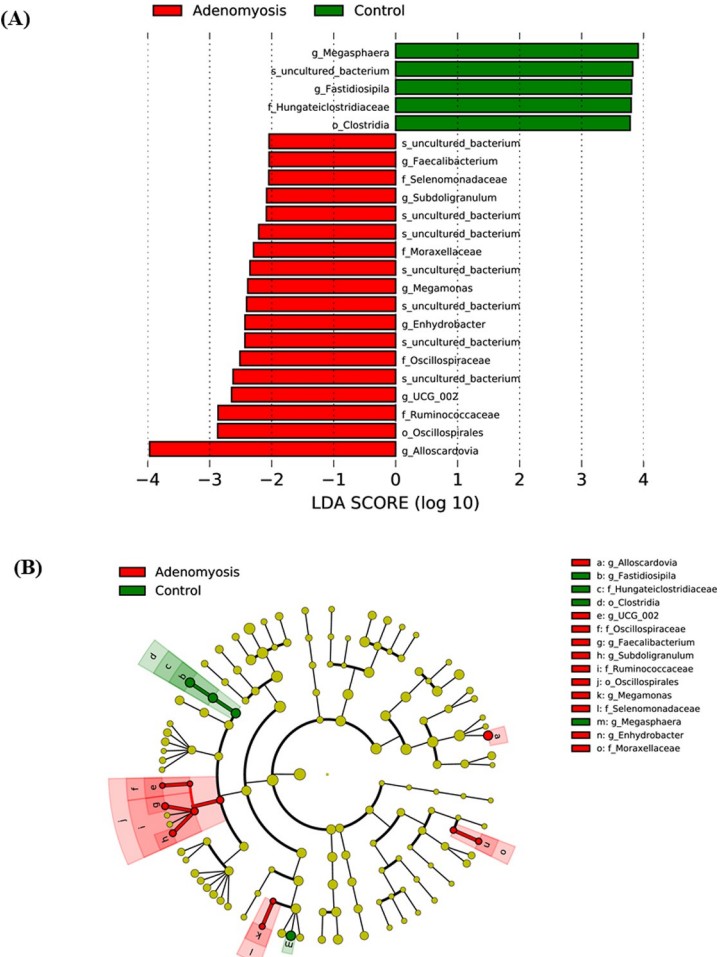

**Fig 4. Linear discriminant analysis effect size (LEfSe) analysis of microbial abundance in the vaginal sample between adenomyosis and normal uterus.** (A) Taxa with a significant difference in both groups were detected by LEfSe analysis with a linear discriminant analysis (LDA) threshold score of 2.0 and a significance of 0.05. (B) Cladogram of detected taxa for each group. Control and adenomyosis taxa are indicated with a positive LDA score (green) and a negative score (red), respectively.

**Table 2. Prevalence of communities in adenomyosis and control groups.**

| Group | CST3 | CST4 | Other | No Type | Total | *P*-value |
|---|---|---|---|---|---|---|
| | n (%) | n (%) | n (%) | n (%) | n (%) | |
| Adenomyosis | 16 (40.0) | 16 (40.0) | 8 (20.0) | 0 (0) | 40 (100.0) | 0.18 |
| Control | 12 (31.6) | 13 (34.2) | 9 (23.7) | 4 (10.5) | 38 (100.0) | |

Note: CST; Community state type. Statistical analysis was performed by the chi-square test.

## Investigation of community state types (CST)

The samples were grouped by CST, with each CST containing the largest proportion of reads, more than 30%. The CST-I, CST-II, CST-III, and CST-V were dominated by *L. crispatus*, *L. gasseri*, *L. iners*, and *L. jensenii*, respectively, while CST-IV was not dominated by *Lactobacilli*. It was defined by the low amount of *Lactobacilli* spp. and enhancement of bacteria associated with bacterial vaginosis (BABV) [46]. The microbiome in adenomyosis groups was 1.3 and 1.2 times as likely to be dominated by the CST-III and CST-IV; however, it was not significantly different (*p* = 0.18). Additionally, eight and nine samples from adenomyosis and control groups were dominated by *L. helveticus*, which were grouped as CST-Other [46] (Table 2).

## Discussion

The present study elucidated differences in vaginal microbiota characteristics between women with and without adenomyosis. The richness of vaginal microbiota was significantly higher in the adenomyosis group, while the other α- and β-diversities were not significantly different. Also, differences in bacterial abundance among both groups were noted. *Megaspehera*, *Fastidiosipila*, Hungateiclostrsidiaceae, and Clostridia were more frequently found in the vaginal microbiota in the control group. On the other hand, *Alloscardovia*, Oscillospirales, Ruminoccoccaceae, *UCG_002*, Oscillospiraceae, *Enhydrobacter*, *Megamonas*, Moraxellaceae, *Subdoligranulum*, Selenomonadaceae, and *Faecalibacterium* in the vaginal microbiota were significantly higher in the adenomyosis versus the control group. *Lactobacillus* was found to be the most prevalent vaginal microbial in both groups. The *L. iners* dominant CST and CST-IV were the most prevalent bacterial communities in the adenomyosis group, while only CST-IV was the most prevalent community in the control group.

The richness of vaginal microbiota in the adenomyosis group was increased compared to the control group but not the evenness of microbiota. The result was not in line with the study done by Chen S *et al.*, who evaluated vaginal microbiota at posterior fornix comparing between women with adenomyosis, endometriosis, adenomyosis and endometriosis, and healthy women. No significant difference was demonstrated in alpha diversity between all four groups in their study [27]. However, the study by Chao et al. was comparable to the present study. They compared vaginal microbiota of women with endometriosis/adenomyosis (EM/AM)-associated chronic pelvic pain (CPP), women with chronic pelvic pain syndrome without EM/AM, and women without CPPS (the control group). Alpha diversity was greatest in women with CPP and EM/AM compared to healthy women and women with CPP [36]. The increased richness of vaginal microbiota is unknown as a cause or consequence of the disease. The clinical significance of this finding is not known. Further study would be needed to clarify this point.

Interestingly, the LEfSe analysis found differential abundances of vaginal bacteria between adenomyosis and control group. The analysis of biomarker bacteria by LEfSe in Chen S *et al.*'s study could not demonstrate the difference among the four groups [27]. In contrast, by using

cross-validated random forest models, Chen C *et al*. could differentiate women with adenomyosis from those without according to OTUs [25]. Chen C's study evaluated microbiota colonized along the reproductive tract in normal and adenomyosis patients.

Local immune dysfunction may participate in the pathophysiology of adenomyosis. Mucosal surface is the first line protection of female reproductive tract [47]. Many studies demonstrated leukocyte infiltration, e.g. macrophages and natural killer cells, in the endometrium of women with adenomyosis [48, 49]. Although most of the vaginal bacteria demonstrated in the present study are commensal microorganisms, few of them are pathogenic such as few species of Moraxellaceae, Clostridia and *G. vaginalis* [50, 51]. *G. vaginalis* is a dominant bacterial species present during bacterial vaginosis, a polymicrobial disorder. Its pathogenicity including attachment to vaginal mucosa, biofilm production and pore formation at cell membrane is associated with its virulence toxins or factors [47]. In the present study, *G. vaginalis* was found as the second order bacterial abundance in both adenomyosis and non-adenomyosis groups; however, it was detected with a larger proportion in the adenomyosis. *G. vaginalis* exerts immunosuppressive effect on the vaginal epithelial tissue since it degrades glycogen in the vaginal epithelium [52, 53]. However, immune response to *G. vaginalis* and its pathogenic effect on uterine epithelium remain unknown. Proof of concept experiment is further needed.

The immune dysregulation also occurred in both eutopic and ectopic endometrium in patients with adenomyosis as demonstrated by the elevated expression of immune checkpoint regulator T cell immunoglobulin mucin molecule 3 (TIM-3) /galectin (Gal-9) and the differential expression of RNA methylation [54, 55]. The increased expression of stimulator of interferon gene (STING), an inducer for type I interferon, in the eutopic endometrium of adenomyosis uterus reflected the involvement of host's innate immune response [56]. The more vaginal bacterial diversity may explain the initiation of host's innate immune response in the eutopic endometrium of adenomyosis.

The analysis of bacterial community state between adenomyosis and control group demonstrated that the CST-III and -IV were dominated in adenomyosis while only CST-IV was dominated in the non-adenomyosis group. CST-III is dominated by *L. iners*, whereas CST-IV is not composed of a large number of *Lactobacillus*. Moreover, CST-IV is characterized by various facultative anaerobes, including *Gardnerella*, *Mobiluncus*, *Atopobium*, *Prevotella* and other genera in the order Clostridiales [44, 57]. It also strongly correlates with BV, but it does not always correlate to BV when diagnosed by clinical symptom and microscopic examination [58]. Although it is common to compare the outcome with CST in the published studies, no comparison of CST in the current three published studies related to adenomyosis was revealed [25, 27, 36]. Chen C *et al*. compared only CST along with the reproductive site but not between control and adenomyosis [25]. Previous data reported that CST-III, dominated with *L. iners*, is more likely a dysbiosis microbiota and more in transition to shift to BV-associated microbiome [59]. The present study found a dominance of CST-III in the adenomyosis group, although not statistically significant. An increased sample size would be needed to clarify in the further study.

In this study, the most abundant vaginal microbial, *Lactobacillus*, was found in proportions of 51.20% and 50.54% in both the adenomyosis and control groups, respectively. The dominance of *Lactobacillus* was in line with several previous studies in which it was the major resident within the vaginal microbial community [12, 25, 60] and also with another study carried out in Thailand, which showed Lactobacilli dominant vaginal microbiota of 56% among 25 healthy participants [61]. Four species of Lactobacilli, *L. iners*, *L. helveticus*, *L. hamsteri*, and *L. vaginalis* were found in the vagina of women in the present study. *L. iners* was the most abundant *Lactobacilli* in both groups (adenomyosis *vs*. control, 37% *vs*. 31%), which was comparable to the previous Thai study [61]. Somewhat unexpectedly, the present study found *L. helveticus* as the second abundance of *Lactobacilli*. We could not explain its existence. After

grouping of vaginal CST by categorizing *L. helveticus* to CST-other, the control and adenomyosis women were dominated with CST-IV and CST-III/CST-IV, respectively. The differences in vaginal bacterial communities found in different studies could be attributed to many factors. First, race and ethnicity could be important factors. Previous data reported that Asian and European ancestors were plausible to have CST-I vaginal microbiome, dominated by *L. crispatus*, while African women were possible to have CST-III, dominated by *L. iners* or *G.vaginalis*. While *L. crispatus* is commonly present in the vagina of healthy women, *L. iners* is found in the vagina of women with either healthy or with BV [62, 63]. CST-IV was common in Hispanic and black women [44]. Although CST found in healthy Japanese women was close to those of white and black North American women, there were some differences in vaginal communities. The frequency of CST-IV was higher in Japanese women than in white women but lower than in black women [64]. Therefore, CST of the vagina in Japanese women may be closer to Thai women. Second, the variation of sample collection sites could impact the differences in vaginal microbiota, e.g., posterior fornix or lower vagina, as demonstrated in the study of Chen C *et al.* [25]. Third, the methods used in each study, such as culture-based or sequencing-based, could attribute to the different vaginal microbiota. Moreover, in the case of gene sequencing, the lack of standardization of analysis software and microbial database results in a unique method for each laboratory with a consequence of the inaccurate comparison and sometimes impossible to compare between the studies.

This study was the pioneer study of vaginal microbiome among women with adenomyosis. In this study, a high-throughput sequencing technique was used to perform microbial analysis, which allowed for the characterization of the in-depth population of molecular vaginal microbiology. This method is a highly effective technique that provides a significantly broader range of microbial diversity compared to the traditional culture-based techniques. In addition, a strict sample collection protocol, including sterile technique, to precisely collect samples from the upper vaginal mucosa without contamination was used in this study. It should be noted that the criteria for adenomyosis diagnosis in this study were based on ultrasound imaging criteria rather than pathological diagnosis. In addition, this study focused only on the genomic properties of the vaginal microbiota between women with and without adenomyosis. Other omics techniques that might have revealed altered mechanisms between the two groups, such as transcriptomic analysis, were not used.

In summary, in this study, an increase in microbial richness were found to correlate with adenomyosis. The different microbiome profiles were noted between adenomyosis and non-adenomyosis group. Mechanisms behind this correlation should now be further investigated as this knowledge would be essential for subsequent clinical implications.

## Supporting information

**S1 Fig. Alpha rarefaction curve of included samples shows saturation of bacterial community with adequate depth and highly abundant microbial diversity.**
(TIF)

**S2 Fig.** Beta diversity analysis of vaginal microbial in women with and without adenomyosis showed no significant difference by A) Bray-Curtis dissimilarities (abundant weighted distance) and B) Jaccard index (the presence or absence of operational taxonomic units [OTUs]).
(TIF)

**S1 Table. Symptoms, characteristics of adenomyosis and other types of gynecological diseases in the adenomyosis group.**
(DOCX)

**S1 File. Data for** Table 1 **&** S1 File**.**
(XLSX)

**S2 File. Phylum-level operational taxonomic units of vaginal microbiota.**
(XLSX)

**S3 File. Genus-level operational taxonomic units of vaginal microbiota.**
(XLSX)

**S4 File. Species-level operational taxonomic units of vaginal microbiota.**
(XLSX)

**S5 File. CST assignment.**
(XLSX)

**S6 File.**
(TXT)

## Acknowledgments

We would like to thank the Center of Excellence in Systems Biology (CUSB), Chulalongkorn University for supporting the high performance computing (HPC) clusters utilized in metagenomic data analysis.

## Author Contributions

**Conceptualization:** Jitsupa Kunaseth, Sunchai Payungporn, Areepan Sophonsritsuk.

**Data curation:** Jitsupa Kunaseth, Wanwisa Waiyaput.

**Formal analysis:** Vorthon Sawaswong.

**Investigation:** Jitsupa Kunaseth, Wanwisa Waiyaput, Prangwalai Chanchaem.

**Methodology:** Jitsupa Kunaseth, Wanwisa Waiyaput, Prangwalai Chanchaem, Rattana Permpech.

**Resources:** Areepan Sophonsritsuk.

**Software:** Vorthon Sawaswong.

**Supervision:** Sunchai Payungporn, Areepan Sophonsritsuk.

**Validation:** Prangwalai Chanchaem.

**Visualization:** Vorthon Sawaswong.

**Writing – original draft:** Jitsupa Kunaseth.

**Writing – review & editing:** Sunchai Payungporn, Areepan Sophonsritsuk.

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
