## [Decision Letter · Decision Letter 0]

13 Aug 2021

PONE-D-21-20218

Vaginal Microbiome of Women with Adenomyosis: A case-control study

PLOS ONE

Dear Dr. Sophonsritsuk,

Thank you for submitting your manuscript to PLOS ONE. After careful consideration, we feel that it has merit but does not fully meet PLOS ONE’s publication criteria as it currently stands. Therefore, we invite you to submit a revised version of the manuscript that addresses the points raised during the review process.

While the reviewers agree that the study has merit, significant issues with the microbiome analysis and data availability must be corrected to make the article suitable for publication.

We look forward to receiving your revised manuscript.

Kind regards,

Christopher Staley, Ph.D.

Academic Editor

PLOS ONE

Journal Requirements:

4. In the ethics statement in the Methods and online submission information, please ensure that you have specified the type of informed consent obtained from the participants (ie written).

6. Thank you for stating the following financial disclosure:

“The authors received no specific funding for this work.”

7. Thank you for stating the following in the Acknowledgments Section of your manuscript:

“The study was financially supported by the Intervention Research Fund of Faculty of Medicine, Ramathibodi Hospital, Mahidol University (project number RF_63085).”

“The authors received no specific funding for this work.”

Reviewers' comments:

Reviewer's Responses to Questions

**Comments to the Author**

1. Is the manuscript technically sound, and do the data support the conclusions?

Reviewer #1: Yes

Reviewer #2: Partly

2. Has the statistical analysis been performed appropriately and rigorously? 

Reviewer #1: Yes

Reviewer #2: No

3. Have the authors made all data underlying the findings in their manuscript fully available?

Reviewer #1: Yes

Reviewer #2: No

4. Is the manuscript presented in an intelligible fashion and written in standard English?

Reviewer #1: Yes

Reviewer #2: Yes

5. Review Comments to the Author

Reviewer #1: The study is original and the experiments, statistics, and other analyses are performed to a high technical standard and are described in sufficient detail. I think the discussion and conclusion can be improved. And the paragraph (lines 296-303) needs to done again, read Chen's study again.

Reviewer #2: The authors conducted an interesting study to evaluate differences in the vaginal microbiome of women with and without adenomyosis. The authors had well phenotyped cases, although the analysis of the vaginal microbiome was superficial relative to other vaginal microbiome studies. The authors only presented aggregate results and did not include any species level differences. This is not worthy given that some Lactobacillus species are thought to be more protective of reproductive health than others, notably L. crispatus. I think the study would be more comparable to other studies evaluating the vaginal microbiome with more complete bioinformatic processing of the sequencing data. There were also some concerns with some of the statements made throughout the manuscript. Some specific concerns are listed below.

This statement on page 6 line 92 needs to be re-worded or clarified:

"Lactobacillus that have been detected in the vagina, only one or two species are dominant in normal host vagina [13]."

The sentence reads like only one or 2 species of Lactobacillus are present ever in the vagina instead of that it is common for women to have their vaginal microbiome composed of largely one lactobacillus species.

Line 96, change invader to pathogens.

Please clarify this statement pg 7 line 101, the meaning is unclear:

“The healthy vaginal flora combines numerous bacterial species in valid proportion with Lactobacilli”

Pg 7 line 103, what do you mean by dysbiosis? Non-lactobacillus predominance? Different from the women’s original baseline microbiome? Some women have diverse microbiota with no known pathology or symptoms.

Please clarify which study you are referring to here pg 8 line 115-117, there is no citation & the Chen study the microbiomes were so different for these women, that women with the condition could be identified by the microbiome from any site along the reproductive tract. Is this the study (ref 22 in your list) to which you are referring? : “Although a previous study reported alterations of the bacterial microbiome in the reproductive tract of adenomyosis women, it focused directly on the correlation between vaginal microbiota composition and adenomyosis and was very limited.”

Why did the authors use Greengenes to classify the sequences? That database is no longer maintained and hasn’t been updated since 2013. Assigning taxonomy with RDP, or another database that is still maintained would be more accurate. Additionally, why were the analysis primarily completed at the phylum and genus level? Given that species level differences are important related to vaginal health outcomes, this should be done and has been consistently done in recent vaginal microbiome literature – particularly speciation of Lactobacillus. Some examples for reference:

Brooks JP, Buck GA, Chen G, et al. Changes in vaginal community state types reflect major shifts in the microbiome. Microb Ecol Health Dis. 2017;28(1):1303265. doi:10.1080/16512235.2017.1303265

Fettweis JM, Serrano MG, Brooks JLPL, et al. The vaginal microbiome and preterm birth. Nat Med. 2019;25(6):1012-1021. doi:10.1038/s41591-019-0450-2

Elovitz MA, Gajer P, Riis V, et al. Cervicovaginal microbiota and local immune response modulate the risk of spontaneous preterm delivery. Nat Commun. 2019;10(1):1305. doi:10.1038/s41467-019-09285-9

Megasphaera species are frequently present in women with bacterial vaginosis and diverse microbiota (versus lactobacillus dominant). It is interesting that this genus was more common in the control group and some discussion on this would be warranted/interesting, since the presence of these microbes has been associated with poor reproductive health outcomes.

Glascock AL, Jimenez NR, Boundy S, et al. Unique roles of vaginal Megasphaera phylotypes in reproductive health. bioRxiv. August 2020:2020.08.18.246819. doi:10.1101/2020.08.18.246819

Why was Lactobacillus dominant classified in this way (pg 14 line 268)? This is not a standard way of classifying or evaluating the vaginal microbiome, which limits the ability to compare results to other studies.

The paragraph beginning at line 304 on pg 16 seems out of place and grossly overstates the results. The authors results presented did not show women with adenomyosis have vaginal dysbiosis (lines 315-318), nor did they present any data indicating these women have inflammation.

Are the data publicly available? The authors said the information is included with the supplementary information, but there is no data. The analysis can’t be reproduced or validated as submitted.

This citation is inaccurate, pg 14 line 264-265: “However, previous reports on the relationship between ethnicity and vaginal Lactobacillus abundance have been inconclusive [40-43].” The studies cited that there are differences in vaginal microbiome communities between ethnicities, although the reason for these differences are unknown. These differences are some of the most reproducible differences observed in the microbiome literature.

There is no figure 1?

6. PLOS authors have the option to publish the peer review history of their article (what does this mean?). If published, this will include your full peer review and any attached files.

Reviewer #1: **Yes: **Bruna Cestari de Azevedo

Reviewer #2: No

---

## [Author Response · Author response to Decision Letter 0]

24 Sep 2021

Review Comments to the Author

Reviewer #1: The study is original and the experiments, statistics, and other analyses are performed to a high technical standard and are described in sufficient detail. I think the discussion and conclusion can be improved. And the paragraph (lines 296-303) needs to done again, read Chen's study again.

Our response:

Thank you so much for looking over our manuscript and gave the informative suggestions.

We are sorry for the mistake.

We went through the Chen C and Chen S studies. We re-wrote the sentences as shown in page 15 line 295-302. (yellow highlight)

Reviewer #2: The authors conducted an interesting study to evaluate differences in the vaginal microbiome of women with and without adenomyosis. The authors had well phenotyped cases, although the analysis of the vaginal microbiome was superficial relative to other vaginal microbiome studies. The authors only presented aggregate results and did not include any species level differences. This is not worthy given that some Lactobacillus species are thought to be more protective of reproductive health than others, notably L. crispatus. I think the study would be more comparable to other studies evaluating the vaginal microbiome with more complete bioinformatic processing of the sequencing data.

Our response:

Thank you so much for reviewing our manuscript and suggest us many good idea.

We agree with you. We re-analyzed the data which included using other database and also species analysis. We also compared the outcomes with other papers as suggestion.

There were also some concerns with some of the statements made throughout the manuscript. Some specific concerns are listed below.

This statement on page 6 line 92 needs to be re-worded or clarified:

"Lactobacillus that have been detected in the vagina, only one or two species are dominant in normal host vagina [13]." The sentence reads like only one or 2 species of Lactobacillus are present ever in the vagina instead of that it is common for women to have their vaginal microbiome composed of largely one lactobacillus species.

Our response:

We made a correction as suggested on page 6, line 93-96. 

 Line 96, change invader to pathogens.

Our response:

We made a correction as suggested on page 6, line 98. 

Please clarify this statement pg 7 line 101, the meaning is unclear:

“The healthy vaginal flora combines numerous bacterial species in valid proportion with Lactobacilli”

Our response:

We made a correction as suggested on page 6 line 103. “The healthy vaginal flora combines numerous bacterial species with a certain large proportion of Lactobacilli”

Pg 7 line 103, what do you mean by dysbiosis? Non-lactobacillus predominance? Different from the women’s original baseline microbiome? Some women have diverse microbiota with no known pathology or symptoms.

Our response:

Thank you so much for this knowledge. 

We made a correction on page 6 line 106. “Alteration of vaginal microbiota”

Please clarify which study you are referring to here pg 8 line 115-117, there is no citation & the Chen study the microbiomes were so different for these women, that women with the condition could be identified by the microbiome from any site along the reproductive tract. Is this the study (ref 22 in your list) to which you are referring? : “Although a previous study reported alterations of the bacterial microbiome in the reproductive tract of adenomyosis women, it focused directly on the correlation between vaginal microbiota composition and adenomyosis and was very limited.”

Our response:

We are sorry for the confusion. We rewrote and cited 3 papers, Chen C, Chen S and Chao on page

7 line 117-122. 

Why did the authors use Greengenes to classify the sequences? That database is no longer maintained and hasn’t been updated since 2013. Assigning taxonomy with RDP, or another database that is still maintained would be more accurate.

Additionally, why were the analysis primarily completed at the phylum and genus level? Given that species level differences are important related to vaginal health outcomes, this should be done and has been consistently done in recent vaginal microbiome literature – particularly speciation of Lactobacillus. Some examples for reference:

Brooks JP, Buck GA, Chen G, et al. Changes in vaginal community state types reflect major shifts in the microbiome. Microb Ecol Health Dis. 2017;28(1):1303265. doi:10.1080/16512235.2017.1303265

Fettweis JM, Serrano MG, Brooks JLPL, et al. The vaginal microbiome and preterm birth. Nat Med. 2019;25(6):1012-1021. doi:10.1038/s41591-019-0450-

Elovitz MA, Gajer P, Riis V, et al. Cervicovaginal microbiota and local immune response modulate the risk of spontaneous preterm delivery. Nat Commun. 2019;10(1):1305. doi:10.1038/s41467-019-09285-9

Our response:

Thank you for your suggestions. We agreed with reviewer’s concern. Therefore, our analysis was improved by changing the database to SILVA 138.1. We decide to use SILVA database because it is larger than Greengenes and RDP. In addition, it was corrected for taxonomy and quality which because it might be more reliable than sequences from NCBI. The results showed the better resolution that most OTUs can be classified at genus level. In addition, it can improve the classification to species level for some OTUs. 

Our analysis mainly focused on the only phylum and genus level because most of OTUs can be clearly classified to genus level due to the limitation of short-read 16S sequencing. [Jeong J, Yun K, Mun S, Chung WH, Choi SY, Nam YD, Lim MY, Hong CP, Park C, Ahn YJ, Han K. The effect of taxonomic classification by full-length 16S rRNA sequencing with a synthetic long-read technology. Sci Rep. 2021 Jan 18;11(1):1727. doi: 10.1038/s41598-020-80826-9.] 

As reviewer suggestion, we adapted the bioinformatic methods to improve the classification of Lactobacillus species. Briefly, we extracted the reads of Lactobacillus and re-classified them by sklearn classifier against custom reference database of Lactobacillus species. 

Megasphaera species are frequently present in women with bacterial vaginosis and diverse microbiota (versus lactobacillus dominant). It is interesting that this genus was more common in the control group and some discussion on this would be warranted/interesting, since the presence of these microbes has been associated with poor reproductive health outcomes.

Glascock AL, Jimenez NR, Boundy S, et al. Unique roles of vaginal Megasphaera phylotypes in reproductive health. bioRxiv. August 2020:2020.08.18.246819. doi:10.1101/2020.08.18.246819

Our response:

Thank you for your suggestions. We discussed about it as suggestion as shown in page 15, line 292-294, 306-308.

Why was Lactobacillus dominant classified in this way (pg 14 line 268)? This is not a standard way of classifying or evaluating the vaginal microbiome, which limits the ability to compare results to other studies.

Our response:

We agree with the reviewer. We correct the paragraph on page 16 line 325-328.

The paragraph beginning at line 304 on pg 16 seems out of place and grossly overstates the results. The authors results presented did not show women with adenomyosis have vaginal dysbiosis (lines 315-318), nor did they present any data indicating these women have inflammation.

Our response:

We agree with the reviewer. We misunderstood. Since we added discussion more for other issues and it might be redundant for having this paragraph, so we decide to delete the whole paragraph.

Are the data publicly available? The authors said the information is included with the supplementary information, but there is no data. The analysis can’t be reproduced or validated as submitted.

Our response:

We are so sorry. We uploaded it this time.

This citation is inaccurate, pg 14 line 264-265: “However, previous reports on the relationship between ethnicity and vaginal Lactobacillus abundance have been inconclusive [40-43].” The studies cited that there are differences in vaginal microbiome communities between ethnicities, although the reason for these differences are unknown. These differences are some of the most reproducible differences observed in the microbiome literature.

Our response:

We discussed more about the ethnicity and vaginal microbiome communities on page 16-17, line 325-353.

There is no figure 1?

Our response:

We are so sorry. It was a mistake while copying the figure.

.

Best regards,

Areepan Sophonsritsuk

---

## [Decision Letter · Decision Letter 1]

12 Nov 2021

PONE-D-21-20218R1Vaginal Microbiome of Women with Adenomyosis: A Case-control StudyPLOS ONE

Dear Dr. Sophonsritsuk,

Thank you for submitting your manuscript to PLOS ONE. After careful consideration, we feel that it has merit but does not fully meet PLOS ONE’s publication criteria as it currently stands. Therefore, we invite you to submit a revised version of the manuscript that addresses the points raised during the review process. As noted by the reviewers, editing for language and further consideration of the discussion are requested.

We look forward to receiving your revised manuscript.

Kind regards,

Christopher Staley, Ph.D.

Academic Editor

PLOS ONE

Journal Requirements:

Reviewers' comments:

Reviewer's Responses to Questions

**Comments to the Author**

1. If the authors have adequately addressed your comments raised in a previous round of review and you feel that this manuscript is now acceptable for publication, you may indicate that here to bypass the “Comments to the Author” section, enter your conflict of interest statement in the “Confidential to Editor” section, and submit your "Accept" recommendation.

Reviewer #1: (No Response)

Reviewer #2: All comments have been addressed

2. Is the manuscript technically sound, and do the data support the conclusions?

Reviewer #1: Yes

Reviewer #2: Yes

3. Has the statistical analysis been performed appropriately and rigorously? 

Reviewer #1: Yes

Reviewer #2: Yes

4. Have the authors made all data underlying the findings in their manuscript fully available?

Reviewer #1: Yes

Reviewer #2: No

5. Is the manuscript presented in an intelligible fashion and written in standard English?

Reviewer #1: Yes

Reviewer #2: Yes

6. Review Comments to the Author

Reviewer #1: I still think that the discussion can be improved, is missing a discussion of concepts between the disease and the results. The authors only can write: "is not known"; "is unknown as a cause or consequence of the disease"; "is unclear"; "we could not explain its existence". What is the relationship between immune dysregulation, adenomyosis and microbiota (results)? The authors can try to explain with concepts (literature) and the results.

Reviewer #2: Thank you for addressing my concerns, I think the manuscript is much improved. I think one of the supporting data files has still not been uploaded (e.g., the information used for tables 1 and S1, nor CST assignments made for individuals). There are some places with some awkward/unusual phrasing, and I think a read through with an editor could improve these. Last request, please change the language throughout (e.g., abstract, intro, conclusion) that states a shift in microbiome is associated with adenomyosis. This was a cross-sectional study, without repeated measures and not change in microbiome over time was analyzed. The analysis conducted in this study does not support a shift, just that there may be different taxas present in people with vs without adenomyosis.

7. PLOS authors have the option to publish the peer review history of their article (what does this mean?). If published, this will include your full peer review and any attached files.

Reviewer #1: No

Reviewer #2: No

---

## [Author Response · Author response to Decision Letter 1]

24 Dec 2021

December 12th , 2021

Dear Professors,

Editors-in-Chief, 

Thank you very much. We deeply appreciate for considering our paper entitled “Vaginal Microbiome of Women with Adenomyosis: A Case-Control Study.” We have made all corrections as suggested by the reviewer and editor. The corrections are as follow in the manuscripts according to our responses to the editor’s comments as listed below.

Review Comments to the Author

Reviewer #1: I still think that the discussion can be improved, is missing a discussion of concepts between the disease and the results. The authors only can write: "is not known"; "is unknown as a cause or consequence of the disease"; "is unclear"; "we could not explain its existence". What is the relationship between immune dysregulation, adenomyosis and microbiota (results)? The authors can try to explain with concepts (literature) and the results.

Our response:

Thank you so much for looking over our manuscript. We appreciate for your suggestion. 

We tried to explain the concepts between the disease and the results as shown in page 15 line 296 to page 16 line 317. (yellow highlight)

Reviewer #2: I think the manuscript is much improved. I think one of the supporting data files has still not been uploaded (e.g., the information used for tables 1 and S1, nor CST assignments made for individuals). There are some places with some awkward/unusual phrasing, and I think a read through with an editor could improve these. Last request, please change the language throughout (e.g., abstract, intro, conclusion) that states a shift in microbiome is associated with adenomyosis. This was a cross-sectional study, without repeated measures and not change in microbiome over time was analyzed. The analysis conducted in this study does not support a shift, just that there may be different taxas present in people with vs without adenomyosis.

Our response:

Thank you so much for reviewing our manuscript again. We appreciate for your suggestion. 

We uploaded the information used for tables 1 and S1 (excel file “Table 1 & Suppl Table 1), and CST assignments (excel file “CST”).

We deleted some awkward paragraph/sentence.

We changed the language throughout (e.g., abstract, intro, conclusion) that states a shift in microbiome is associated with adenomyosis as suggestion on page 3, line 53-55, page 7, line 127-128, and page 18, line 374-375. 

.

Best regards,

Areepan Sophonsritsuk

---

## [Decision Letter · Decision Letter 2]

18 Jan 2022

Vaginal Microbiome of Women with Adenomyosis: A Case-control Study

PONE-D-21-20218R2

Dear Dr. Sophonsritsuk,

We’re pleased to inform you that your manuscript has been judged scientifically suitable for publication and will be formally accepted for publication once it meets all outstanding technical requirements.

Kind regards,

Christopher Staley, Ph.D.

Academic Editor

PLOS ONE

Additional Editor Comments (optional):

Reviewers' comments:

Reviewer's Responses to Questions

**Comments to the Author**

1. If the authors have adequately addressed your comments raised in a previous round of review and you feel that this manuscript is now acceptable for publication, you may indicate that here to bypass the “Comments to the Author” section, enter your conflict of interest statement in the “Confidential to Editor” section, and submit your "Accept" recommendation.

Reviewer #1: (No Response)

Reviewer #2: All comments have been addressed

Reviewer #3: All comments have been addressed

2. Is the manuscript technically sound, and do the data support the conclusions?

Reviewer #1: Partly

Reviewer #2: Yes

Reviewer #3: Yes

3. Has the statistical analysis been performed appropriately and rigorously? 

Reviewer #1: Yes

Reviewer #2: Yes

Reviewer #3: I Don't Know

4. Have the authors made all data underlying the findings in their manuscript fully available?

Reviewer #1: Yes

Reviewer #2: Yes

Reviewer #3: Yes

5. Is the manuscript presented in an intelligible fashion and written in standard English?

Reviewer #1: Yes

Reviewer #2: Yes

Reviewer #3: Yes

6. Review Comments to the Author

Reviewer #1: The author did not know how to make a consistent discussion. The data is interesting, but you have to know how to use this data for a good discussion.

Reviewer #2: The authors have addressed my comments from the previous reviews, I have no other comments for this paper.

Reviewer #3: well done study, much improved after addressing reviewers comments. It is an original and the methodology are performed to a high technical standard and are described in sufficient detail.

7. PLOS authors have the option to publish the peer review history of their article (what does this mean?). If published, this will include your full peer review and any attached files.

Reviewer #1: **Yes: **Bruna Cestari de Azevedo

Reviewer #2: No

Reviewer #3: **Yes: **Aboubakr Mohamed Elnashar

---

## [Editor Report · Acceptance letter]

3 Feb 2022

PONE-D-21-20218R2 

Vaginal Microbiome of Women with Adenomyosis: A Case-control Study 

Dear Dr. Sophonsritsuk:

I'm pleased to inform you that your manuscript has been deemed suitable for publication in PLOS ONE. Congratulations! Your manuscript is now with our production department. 

Kind regards, 

on behalf of

Dr. Christopher Staley 

Academic Editor

PLOS ONE